# Evaluation of Endocan as a Treatment for Acute Inflammatory Respiratory Failure

**DOI:** 10.3390/cells12020257

**Published:** 2023-01-07

**Authors:** Maxence Hureau, Lucie Portier, Méline Prin, Patricia de Nadai, Joanne Balsamelli, Anne Tsicopoulos, Daniel Mathieu, Philippe Lassalle, Bogdan Grigoriu, Alexandre Gaudet, Nathalie De Freitas Caires

**Affiliations:** 1Univ. de Lille, CNRS, Inserm, CHU Lille, Institut Pasteur de Lille, U1019-UMR9017-CIIL-Centre d’Infection et d’Immunité de Lille, CHU Lille, Surgical Critical Care, Department of Anesthesiology and Critical Care, F-59000 Lille, France; 2Biothelis, F-59000 Lille, France; 3Centre Hospitalier de Valenciennes, Laboratoire d’Anatomopathologie, F-59300 Valenciennes, France; 4Univ. Lille, CNRS, Inserm, CHU Lille, Institut Pasteur de Lille, U1019-UMR9017-CIIL-Centre d’Infection et d’Immunité de Lille, F-59000 Lille, France; 5Univ. Lille, CNRS, Inserm, CHU Lille, Institut Pasteur de Lille, U1019-UMR9017-CIIL-Centre d’Infection et d’Immunité de Lille, CHU Lille, Service de Pneumologie et Immuno-Allergologie, Centre de Compétence pour les Maladies Pulmonaires Rares, F-59000 Lille, France; 6Univ. Lille, CNRS, Inserm, CHU Lille, Institut Pasteur de Lille, U1019-UMR9017-CIIL-Centre d’Infection et d’Immunité de Lille, CHU Lille, Pôle de Médecine Intensive—Réanimation, F-59000 Lille, France; 7Service des Soins Intensifs et Urgences Oncologiques, Institut Jules Bordet, Université Libre de Bruxelles (ULB), 1050 Brussels, Belgium

**Keywords:** acute respiratory distress syndrome, acute lung injury, respiratory failure, endocan, Esm-1

## Abstract

Background: Acute respiratory distress syndrome (ARDS) is a life-threatening condition resulting from acute pulmonary inflammation. However, no specific treatment for ARDS has yet been developed. Previous findings suggest that lung injuries related to ARDS could be regulated by endocan (Esm-1). The aim of this study was to evaluate the potential efficiency of endocan in the treatment of ARDS. Methods: We first compared the features of acute pulmonary inflammation and the severity of hypoxemia in a tracheal LPS-induced acute lung injury (ALI) model performed in knockout (*Esm1*^−/−^) and wild type (WT) littermate C57Bl/6 mice. Next, we assessed the effects of a continuous infusion of glycosylated murine endocan in our ALI model in *Esm1*^−/−^ mice. Results: In our ALI model, we report higher alveolar leukocytes (*p* < 0.001), neutrophils (*p* < 0.001), and MPO (*p* < 0.001), and lower blood oxygenation (*p* < 0.001) in *Esm1*^−/−^ mice compared to WT mice. Continuous delivery of glycosylated murine endocan after LPS-induced ALI resulted in decreased alveolar leukocytes (*p* = 0.012) and neutrophils (*p* = 0.012), higher blood oxygenation levels (*p* < 0.001), and reduced histological lung injury (*p* = 0.04), compared to mice treated with PBS. Conclusions: Endocan appears to be an effective treatment in an ARDS-like model in C57Bl/6 mice.

## 1. Introduction

Acute respiratory distress syndrome (ARDS), an acute lung inflammation responsible of severe hypoxemia, is a common and lethal condition in critically ill patients [1]. Unbalanced lung inflammation following direct pulmonary aggression, such as bacterial pneumonia or COVID-19 infection, is a major contributor to ARDS. The major histological feature of ARDS, known as diffuse alveolar damage (DAD), may result from endothelial and epithelial injury, neutrophil-dependent lung invasion, and acute systemic pro-inflammatory states [2]. The migration of leukocytes, and especially neutrophils, from the blood compartment to the alveolar space plays a key role in the development of DAD [3]. Pre-emptive therapeutics to decrease leukocyte diapedesis could alleviate DAD and improve the prognosis of ARDS [1].

Endocan (Esm-1) is a 50 kilodaltons (kDa) proteoglycan mainly expressed by lung endothelial cells and upregulated by pro-inflammatory cytokines [4]. Previous laboratory data have shown that endocan could interfere with ICAM-1-dependent adhesion and the migration of human and mouse activated leukocytes, thus inhibiting their trans-endothelial migration. In addition, the protective effect of endocan on LPS-induced acute lung injury has been reported in Balb/c mice in a previous study [5]. However, the conclusions to be drawn from these data were limited, since major interference with the endogenous murine endocan could not be ruled out. Indeed, mouse endocan seems to be characterized by the co-existence of two functionally opposite forms: one glycosylated, with likely anti-inflammatory effects, and another non-glycosylated, which is thought to act as an antagonist of glycosylated endocan [6,7]. Subsequently, further models allowing for the isolation of the effects of glycosylated endocan was deemed necessary.

Therefore, the aim of our study was to assess the hypothetical anti-inflammatory effect of the glycosylated form of endocan, which is the only type found in humans.

Several clinical studies conducted in trauma and septic patients support the hypothesis of an anti-inflammatory role of endocan, as higher blood concentrations of endocan at admission in intensive care units seem to reduce the risk of further progression to ARDS [8,9,10]. Therefore, it has been proposed that a lack of secretion of endocan could be associated with a higher risk of respiratory failure and ARDS [7]. 

The aim of this study was to assess the anti-inflammatory effect of endocan, while ruling out any possible interference of endogenous endocan in our explorations. Furthermore, the therapeutic effect of endocan on hypoxemia, reflecting the severity of respiratory failure, has never been explored. To these ends, we compared the features of lung inflammation as well as the severity of hypoxemia in an LPS-induced ALI model conducted in endocan knockout (*Esm1*^−/−^) and WT littermate mice. However, in contrast with human endocan, which is exclusively found in a fully glycosylated form, murine endocan is physiologically found in two forms: a 50 kDa glycosylated type, similar to human endocan, and a 20 kDa non-glycosylated type. Therefore, the results obtained when comparing *Esm1*^−/−^ and WT mice may reflect either the effect of glycosylated or non-glycosylated endocan. We thus aimed to specifically explore, in a complementary model, the effects of a continuous infusion of glycosylated endocan in an LPS-induced ALI model in *Esm1*^−/−^ mice.

## 2. Materials and Methods

### 2.1. Production and Purification of Murine Endocan

We produced and purified murine glycosylated endocan in our laboratory using stably transfected CHO DG44 cells, as previously described [11]. Cells were cultured within CL1000 bioreactors (ref. Z688029, Integra Biosciences, Cergy, France) and in serum-free CHO media (Gibco, Waltham, MA, USA). The supernatant of the cells was enriched with recombinant murine glycosylated endocan. Then, the recombinant protein was purified on an ion exchange column (HiTrap Q FF (ref. 17-5156-01, GE Healthcare, Chicago, IL, USA)), followed by an affinity column coated with the anti-murine endocan MEP14 monoclonal antibody, as previously described [11]. We concentrated the protein by centrifugation using Amicon^®^ Ultra Centrifugal Filters and finally, titrated the protein concentration using the endocan ELISA kit (ref. LIK-1101, DIYEK M1, Biothelis, France). Each production batch of the recombinant protein was controlled using a Western blot assay to confirm that the purified protein only consisted of glycosylated endocan. The preparations were shown to be LPS-free using the LAL test (ref. 50-650U, Lonza, Switzerland). 

### 2.2. Mice and Ethical Aspects

We used C57Bl/6 *Esm1*^−/−^ and WT littermate 7–11-week-old male mice, bred in our animal facility. The study protocol was approved by the local ethics committee (N° APAFIS#6059). All experimentations complied with standard ethical practices.

### 2.3. Generation of Esm1 Knockout Mice 

*Esm1*^−/−^ mice were generated as described elsewhere [12]. Briefly, loxP recognition loci were inserted into flank exons 1 and 2 of the mouse *Esm1* gene (Ensembl ID ENSMUSG00000042379). In collaboration with the Institut Clinique de la Souris (Strasbourg, France), a targeting cDNA fragment harboring a neomycin resistance cassette flanked by loxP loci was transfected into mouse embryonic stem cells by electroporation, resulting in the replacement of *Esm1* exons 1 and 2 with the neomycin resistance cassette by homologous recombination. After transfection, embryonic stem cells for *Esm1* recombination were selected using G418-containing medium, and these clones were validated by Southern blotting and injected into blastocysts and then into pseudopregnant female mice. The progeny was then crossed to generate mice lacking exons 1 and 2 of *Esm1* on homologous chromosomes. These mice were then crossed with Cre deleterious mice to eliminate the neomycin-resistant cassette. For genotyping, we extracted genomic DNA from the mouse ears and performed PCR with the REDExtract-N-Amp™ Tissue PCR kit (ref. R4775, Sigma-Aldrich, St. Louis, MO, USA), following the manufacturer’s instructions, and analyzed the samples on agarose gel. We used primer pairs 1f (ATGTCTATGTCAGTCTCCTC) and 1r (CTCTTGCCAGCCTCTCTTGT) to generate a 310 bp fragment for the wild-type allele and primer pairs 1f and 2r (CTCCAAAATCCAGAAACCC) to generate a 442 bp fragment for the knockout allele. 

### 2.4. Experimental Protocols

The two experimental protocols carried out in our study are described in Figure 1. In the first protocol (Figure 1A), we used C57Bl/6 *Esm1*^−/−^ and WT littermate mice, as previously described. After intratracheal instillation of LPS, the clinical examination and evaluation of physiological pulmonary function were performed each day, as described. The mice were sacrificed at different time points in order to obtain the complete kinetics of each parameter explored from Day 0 to Day 10. In the second protocol, described in Figure 1B, we exclusively used C57Bl/6 *Esm1*^−/−^ mice. We performed subcutaneous implantation of osmotic pumps (ref. 1003D, Alzet, Cupertino, CA, USA) previously filled with PBS or murine endocan, as described, followed by the intratracheal instillation of LPS. A non-lethal blood sample was collected by retro orbital puncture just before the intratracheal instillation. We performed the same follow-up as in the first protocol, except that the mice were sacrificed on Day 5 by the intraperitoneal injection of pentobarbital (100 mg/kg). 

### 2.5. Preparation of Osmotic Pumps and Subcutaneous Implantation

We used osmotic pumps to obtain the continuous delivery of endocan for 3 days at 1 µL/h. Each pump of 100 µL was filled with either 200 µg of murine glycosylated endocan (100 µL of endocan solution at a final concentration of 2 mg/mL) or with 100 µL of PBS. The filled pumps were placed in PBS for a maximal duration of 4 h before subcutaneous implantation. We then anesthetized the mice following the aforementioned protocol, and performed subcutaneous the implantation of the osmotic pumps after antiseptic preparation of the skin. General anesthesia was then initiated using intraperitoneal injection of 5 mg/kg of atipamezole.

### 2.6. Preparation and Administration of LPS for the ALI Model

LPS used in our protocol (E. Coli O111: B4, Sigma-Aldrich, St. Louis, MO, USA) was prepared in sterile conditions. We diluted 10 mg/kg of LPS in PBS for a total volume of 50 µL for each mouse. For the ALI protocol, all mice underwent general anesthesia using an intraperitoneal injection of 75 mg/kg of ketamine and 0.5 mg/kg of medetomidine, followed by the intratracheal instillation of LPS. General anesthesia was then initiated, as previously described.

### 2.7. Clinical Follow-Up and Noninvasive Oximetry

In both protocols, daily clinical examination was performed to assess mice viability and weight. Noninvasive recording of oximetry, heart rate, and respiratory rate were performed daily and averaged over 5 to 10 min on live mice, using the MouseOx^®^ Plus device (Starr Life Sciences Corp, Oakmont, PA, USA).

### 2.8. Biological Sampling after Sacrifice and Conservation

Immediately after sacrifice, we collected blood samples, BAL, and the lungs from the mice. Blood samples were collected by obtaining a section of large thoracic vessels, which were then stored in aliquot at room temperature until coagulation. Then, samples were centrifugated at 1500× *g* for 5 min at room temperature. The serum and centrifugation pellets were separated, and the serum was stored at −20 °C. BAL were performed on the right lung by tracheal cannulation and after ligation of the left bronchus, with two successive washes of 500 μL of sterile PBS, and ice-stored in aliquots. BAL were centrifugated at 1500× *g* for 10 min at 4 °C. The supernatant and the centrifugation pellets were separated, and the supernatant was stored at −20 °C. The centrifugation pellets were then mixed with 1 mL of PBS, and the cells were manually counted using Thoma counting chambers. One hundred thousand cells were spotted using a Shandon Cytospin 4 device (Thermo electron corporation, Waltham, MA, USA), and then colored by May-Grünwald Giemsa. The left lung was placed in Antigenfix^®^ (ref. 2545826, MM FRANCE, France) for 4 h, then rinsed in two successive PBS baths. The lungs were gradually dehydrated in successively increasing alcohol concentration baths, put in Diasolv^®^ (ref. 2545839, MM FRANCE, France) for 30 min, and embedded in paraffin overnight. 

### 2.9. Cell Counting and Histological Examination

For each BAL Cytospin cell identification, at least 200 cells were manually counted by an operator blinded to the mice groups. Each leukocyte visualized was categorized as a neutrophil, macrophage/monocyte, or lymphocyte. The respective proportions and absolute counts were subsequently calculated for each cell type. The paraffin-embedded lungs were prepared for histological examination with microtome to obtain sections of 5 µm thickness. Samples were then stained with hematoxylin and eosin and examined by a trained pathologist blinded to the mouse treatment to determine the main histological features of the mouse acute lung injury using the lung injury scoring system (LISS) At least 20 random fields were independently scored regarding several parameters, including neutrophils in the alveolar space, neutrophils in the interstitial space, hyaline membranes, proteinaceous debris filling the air spaces, and alveolar septal thickening. A lung injury score was then calculated by the addition of each of the five independent parameters previously mentioned, weighted according to their relevance, as described by Matute-Bello, et al. [13].

### 2.10. Evaluation of Total Proteins, MPO, TNFα, and Endocan Concentrations in Sera and in BAL Samples

The sera and BAL samples were thawed and homogenized by soft shaking. The BAL protein concentrations were assessed with a Pierce™ BCA Protein Assay Kit (ref. 23227, Thermo Fisher Scientific, Waltham, MA, USA). The mouse myeloperoxidase (mMPO) concentrations in BAL were assessed with mMPO DuoSet ELISA kit (ref. DY3667, R&D Systems, Minneapolis, MN, USA). The mouse TNF-alpha (mTNF-α) concentrations in BAL were assessed with the mTNF DuoSet ELISA kit (ref. DY410, R&D Systems, Minneapolis, MN, USA). The murine endocan concentrations in the sera were assessed using DIYEK M1 ELISA kit (ref. LIK-1101, Biothelis, Lille, France). 

### 2.11. Statistics

All statistical analyses were performed with SPSS^®^ 26 software (IBM, Armonk, NY, USA) for Windows, and all graphs were created using GraphPad Prism 6.0 software. The normality of quantitative variables was not tested because of the small sample sizes. We used Mann–Whitney tests to compare the distributions of groups of unpaired variables. For the evaluation of the effect of the group in our first protocol, from D0 to D10 for inflammatory parameters and oximetry, as well as oximetry in our second protocol, we used a linear mixed model test with the Satterthwaite approximation using the variable “Group” as the fixed effect and the variable “Time” as the random effect. The distributions of values from a repeated variable were compared using the Kruskal–Wallis tests. All tests were performed with a 2-tailed alpha-risk set at 0.05.

## 3. Results

### 3.1. Endogenous Murine Endocan Decreases the Severity of ALI and Alleviates Lung Physiological Dysfunction

In the first model, we found a significant variation in serum endogenous murine endocan over time in our LPS-induced ALI model (*p* < 0.001) (Figure 2A). The median concentration of murine endocan was measured at 2.59 ng/mL [2.39; 2.78] at D0, significantly decreased at 12 h (H12) (0.42 [0.35; 0.68]) and returned to baseline levels at D10 (2.28 [1.94; 2.57]). 

We observed that global BAL leukocyte recruitment from D0 to D10 after LPS-induced ALI was more pronounced in the *Esm1*^−/−^ mice than in WT mice (*p* < 0.001) (Figure 2B). *Esm1*^−/−^ mice show more leukocyte recruitment at D1, D2, D4, and D5 than WT mice (D1: 775 × 10^3^ cells/mL [700; 1500] vs. 450 [340; 465], *p* < 0.001, D2: 1440 [1130; 1850] vs. 940 [870; 1140], *p* = 0.011, D4: 2280 [1520; 3010] vs. 1365 [985; 2180], *p* = 0.03 and D5: 2270 [2090; 3300] vs. 945 [690; 1160], *p* = 0.004) (Figure 2B). The maximal leukocyte recruitment was delayed at D5 for the *Esm1*^−/−^ mice versus D3 for the WT mice. 

We obtained similar findings for neutrophil recruitment from D0 to D10, which was higher in the *Esm1*^−/−^ than in the WT mice (*p* < 0.001) (Figure 2C). We also noted higher neutrophil recruitment at D1, D2, D4, and D5 in the *Esm1*^−/−^ mice than in the WT mice (D1: 696 × 10^3^ cells/mL [602; 1425] vs. 335 [278; 406], *p* = 0.003, D2: 1117 [1029; 1756] vs. 853 [689; 950], *p* = 0.007, D4: 1932 [1251; 2153] vs. 577 [195; 1765], *p* = 0.02, and D5: 1636 [1407; 1698] vs. 365 [198; 646], *p* = 0.01) (Figure 2C). As for the total cells, the maximal neutrophil recruitment was delayed, being observed at D4 for the *Esm1*^−/−^ mice versus D3 for the WT mice.

We did not find any significant effect of the groups for macrophage recruitment from D0 to D10 (*p* = 0.57) (Figure 2D). The maximum macrophage recruitment was observed at D6 in the *Esm1*^−/−^ mice (536 × 10^3^ cells/mL [358; 753]) and at D5 (687 × 103 cells/mL [620; 940]) in the WT mice. 

The concentration of mMPO in BAL was higher in the *Esm1*^−/−^ mice than in the WT mice (*p* < 0.001) (Figure 2E). We found higher mMPO concentrations in the *Esm1*^−/−^ mice, compared to the WT mice at D5 and D6, respectively (1437 µg/mL [1200; 1657] vs. 547 [441; 1110], *p* = 0.015, and 796 [512; 1070] vs. 211 [191; 417], *p* < 0.001). Maximal mMPO values were found at D5 in the *Esm1*^−/−^ group (1437 µg/mL [1200; 1657]) and at D4 in the WT group (1153 µg/ml [772; 1815]).

We did not find any differences in the alveolar mTNF and protein concentrations between the WT and *Esm1*^−/−^ mice (Figure 2F,G).

Using non-invasive oximetry, we observed a significant worsening in the pulmonary physiological function over time from D0 to D10 in the *Esm1*^−/−^ mice compared to the WT mice (*p* < 0.001) (Figure 2H). We also found lower SpO2 values in the *Esm1*^−/−^ mice at D4 and D5 (respectively for *Esm1*^−/−^ and WT groups: 74% [71; 78] vs. 84% [63; 92], *p* = 0.035 and 69% [64; 70] vs. 81% [61; 85], *p* = 0.017) (Figure 2H). We found no difference in heart and respiratory rates over time from D0 to D10 between the two groups, respectively (*p* = 0.539 and *p* = 0.075).

### 3.2. Endocan Supplementation after LPS-Induced ALI in Esm1^−/−^ Mice Decreases Lung Inflammation and Alleviates Pulmonary Physiological Dysfunction

While the aforementioned results reflect the effects of both forms of murine endogenous endocan (i.e., glycosylated and non-glycosylated endocan), in a second protocol, we aimed to specifically explore the biological role of glycosylated endocan, which is the only form of endocan secreted in humans. To this end, we compared endocan-treated *Esm1*^−/−^ mice to PBS-treated *Esm1*^−/−^ mice, and to WT mice at D5 of LPS-induced ALI.

Serum concentrations of murine endocan at D0 were measured at 67.8 ng/mL [45.9; 85.3] in the *Esm1*^−/−^ mice treated with endocan infusion, and at 2.58 ng/mL [2.39; 2.78] in the WT mice. 

*Esm1*^−/−^ mice treated with endocan showed a significant decrease in total leukocytes, neutrophils, and macrophage recruitment compared to those treated with PBS (respectively, 865 × 10^3^ cells/mL [640; 1130] vs. 2130 [1540; 2550], *p* = 0.012, for total leukocytes; 638 × 10^3^ cells/mL [476; 855] vs. 1489 [1285; 2146], *p* = 0.012, for neutrophils; and 164 × 10^3^ cells/mL [141; 255] vs. 404 [354; 610], *p* = 0.018, for macrophages). In addition, we found significantly higher numbers of macrophages in WT mice than in *Esm1*^−/−^ mice treated with endocan (687 × 10^3^ cells/mL [594; 942] vs. 164 × 10^3^ cells/mL [141; 255], *p* = 0.003 (Figure 3A–C). The protein concentration in BAL was lower in the endocan group than in the PBS group (965 µg/mL [690; 1040] vs. 1417 [1306; 1655], *p* = 0.05) (Figure 3F).

There was no difference in mMPO and mTNF concentrations between the two groups (for endocan and PBS treatments, respectively, 1275 µg/mL [774; 1930] vs. 1840 [1390; 2890], *p* = 0.145, and 207 pg/mL [178; 373] vs. 348 [183; 631], *p* = 0.388) (Figure 3D,E).

The evaluation of physiological pulmonary dysfunction is shown in Figure 3G. Mice treated with endocan exhibited greater SpO2 values than control mice over time from D0 to D5 (*p* < 0.001), with a significant difference at D4 and D5 (respectively 84% [76; 89] vs. 67% [65; 78], *p* = 0.023, at D4, and 78% [73; 90] vs. 69% [63; 73], *p* = 0.02, at D5). There was no difference between the two groups for heart rate and respiratory rate over time from D0 to D5 (respectively *p* = 0.771 and *p* = 0.522). 

### 3.3. Histological Features of LPS-Induced ALI Are Reduced by Endocan Supplementation in Esm1^−/−^ Mice

Regarding histopathological examinations, we observed a significant alleviation of lung injuries, notably driven by a decrease in neutrophilic infiltrates, hyaline membranes, and fibrin deposits in the endocan-treated group (Figure 4A) compared to the control group (Figure 4B). Accordingly, we found lower LISS values in the endocan (0.571 [0.527; 0.732]) than in the PBS-osmotic pump group (0.711 [0.644; 0.794]) (*p* = 0.04) (Figure 4C). However, due to the wide variability between individual animals, there was no significant difference in LISS values between the WT and *Esm1*^−/−^ mice evaluated at 0.578 [0.541; 0.680] vs. 0.756 [0.530; 0.772], *p* = 0.699), respectively (Figure 4C).

## 4. Discussion

ARDS is a severe disease resulting from a dysregulated neutrophilic-dependent lung invasion [2]. Endocan could play an anti-inflammatory role through its direct interaction with the leukocyte integrin LFA-1, subsequently leading to the alleviation of dysregulated leukocyte diapedesis [14]. Data from the literature suggest that the anti-inflammatory action of human endocan relies on both the protein core and the presence of its glycanic chain. As a consequence, the suppression of the glycosylated portion of the endocan seems to be associated with the loss of this anti-inflammatory effect [14]. In contrast with human endocan, which is exclusively found in a fully glycosylated form, murine endocan is physiologically found in two forms: the 50 kDa glycosylated type, similar to human endocan, and the 20 kDa non-glycosylated type. Such a non-glycosylated endocan has been found to be pro-inflammatory in vivo due to its antagonizing effect on the anti-inflammatory influence of glycosylated endocan [15]. Therefore, the endogenous secretion of non-glycosylated endocan may result in significant bias in the interpretation of ALI models in mice [6,15], making it difficult to assess the exact effect of glycosylated endocan. Thus, the aim of this study was to specifically evaluate the physiological effect of glycosylated endocan in an LPS-induced model of ALI, using *Esm1*^−/−^ mice to rule out any potential interference of endogenous endocan. 

In order to compare our results with previously published data, we assessed the inflammatory and histological features of lung inflammation according to the guidelines released by the American Thoracic Society for measurements of experimental acute lung injury in animals [13]. To specifically address the effect of endocan, all C57BL/6 mice used in this study were similar in age and weight, as well as being littermates. The only difference between the groups was the presence or the absence of endocan. Finally, to study the effects of a continuous infusion of endocan, we used osmotic pumps in order to obtain a constant delivery of glycosylated murine endocan over 3 days.

When comparing *Esm1*^−/−^ mice and WT littermates, we found that endogenous murine endocan may potentially decrease the severity of LPS-induced ALI and alleviate lung physiological dysfunction, as reflected by greater SpO2 values in the WT group (Figure 2). However, the LISS values were not different between the two groups (Figure 4). In contrast, we found significant differences in BAL cellularity and SpO2 between the *Esm1*^−/−^ mice and their WT littermates. This discrepancy may be explained by a higher variability in LISS assessment, which does not strictly rely on a numerical result, but also on semi-quantitative evaluations for several parameters.

A model of endocan supplementation in *Esm1*^−/−^ mice was therefore used to establish an in vivo model to assess clearly the physiological role of the glycosylated endocan, which is the unique form in humans. Indeed, the continuous infusion of murine glycosylated endocan in the *Esm1*^−/−^ mice led to an alleviation of lung inflammation associated with an improvement in pulmonary physiological function when compared with the *Esm1*^−/−^ mice treated with PBS. 

Moreover, the continuous infusion of glycosylated murine endocan in the *Esm1*^−/−^ mice reduced the histological features of LPS-induced ALI, as reflected by the lower LISS values observed in the endocan-treated *Esm1*^−/−^ mice than those observed in the PBS-treated WT mice. Such low LISS values were not observed in the BALB/c mice [5], confirming that the endogenous endocan can interfere with the exogenous type. Such interference may explain some conflicting results due to the use of a non-qualified glycosylation form of endocan in experimental models [6].

Taken together, all these data reinforce the hypothesis that glycosylated endocan exerts anti-inflammatory effects, as it is likely to reduce the severity of lung inflammation and improve the pulmonary physiological function after LPS-induced ALI. These data are consistent with those in previously published studies, showing that the injection of human endocan in BALB/c mice alleviates the features of LPS-induced ALI and attenuates the respiratory physiological dysfunction [5,16].

Interestingly, when comparing *Esm1*^−/−^ mice treated with endocan to WT mice, the only significant difference between them was regarding the macrophages, which showed greater counts in WT mice. Importantly, we found no significant difference between the WT mice and the endocan-treated *Esm1*^−/−^ mice for all other variables explored, yet the conclusions to be drawn from these results remain limited, as the balance between glycosylated and non-glycosylated endocan in WT mice cannot be measured.

Our subcutaneous infusion model allowed for the achievement of sustainable high circulating concentrations of endocan, while previously published models failed to maintain such concentrations [5]. However, several issues can be raised: (1) there was a variable diffusion (at equal loaded amounts, blood endocan levels vary from 25 to 116 ng/mL, far greater than values observed in WT mice); (2) we did not measure endocan blood levels beyond Day 0 in the endocan-treated *Esm1*^−/−^ mice, limiting the interpretability of our results, and (3) this remains a preventive model in which endocan starts to spread before the initiation of experimental ARDS. Therefore, much more work is required to refine the significance of endocan administration during the course of ALI.

Additional limitations to this study are: first, it would have been interesting to more precisely evaluate the changes in the M1 and M2 macrophage subsets in our models, yet this characterization was not performed in our study. Secondly, the accurate assessment of changes in vascular density and vascular leakage in endocan-treated mice would have been necessary to understand the changes in immune cell infiltration reported in our results. This was assessed through the measurement of BAL proteins, which remains an imperfect method for the estimation of this parameter. Thirdly, assessing the respective levels of non-glycosylated and glycosylated endocan in mice would have been informative for the interpretation of our results. However, there is no method currently available to perform the quantifications of non-glycosylated and glycosylated endocan. This latter point underlines the importance of our study, which is, to our knowledge, the first to describe an in vivo model able to clearly decipher the role of glycosylated endocan by exploring the action of the continuous infusion of endocan to achieve a supra-physiological concentration. 

## 5. Conclusions

In conclusion, using a model of LPS-induced ALI in C57BL/6 mice, we show that treatment by endocan results in a decrease in lung inflammation, subsequently leading to the attenuation of respiratory failure. Notably, our findings are supported by consistent data obtained after endocan treatment in *Esm1*^−/−^ mice, thereby ruling out interference from glycosylated, and most importantly, non-glycosylated, murine endogenous endocan. 

Therefore, endocan appears as a promising novel therapy for the treatment of ARDS. Further explorations are warranted to confirm these results.

## Figures and Tables

**Figure 1 cells-12-00257-f001:**
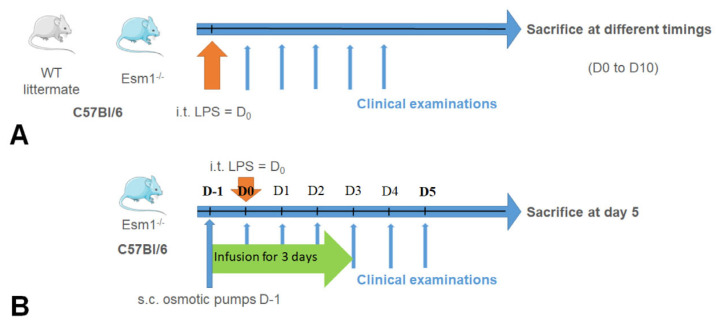
Experimental protocols. The first protocol (**A**) was performed to study differences of the LPS induced ALI between WT littermate and *Esm1*^−/−^ C57Bl/6 mice. The intra tracheal instillation of LPS was performed at D0, the clinical examination, with noninvasive measurement of oximetry, was performed each day, and the mice were sacrificed at different times to describe the features of ALI at each timing. The second protocol (**B**) was performed to study the effect of the continuous infusion of mouse endocan by osmotic pumps in *Esm1*^−/−^ mice. Osmotic pumps were filled with either endocan or PBS and implanted subcutaneously one day before i.t. LPS instillation. Clinical examination, including noninvasive measurement of oximetry, was performed daily, and the mice were sacrificed at day 5 to describe the features of ALI. LPS: lipopolysaccharides; i.t.: intra tracheal instillation.

**Figure 2 cells-12-00257-f002:**
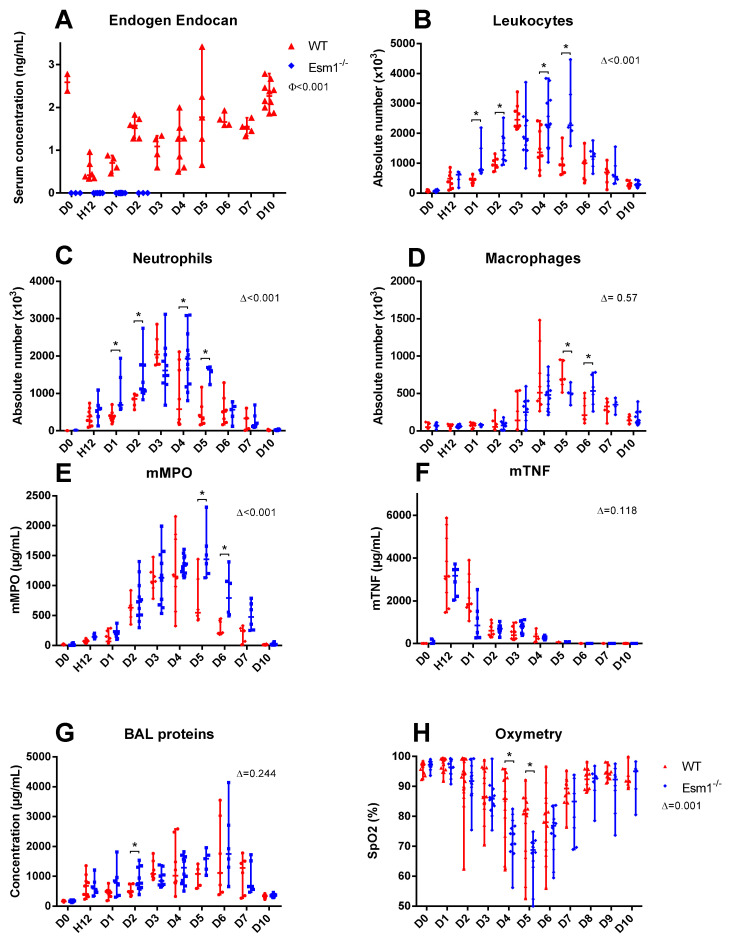
Endogenous endocan alleviates the inflammatory response and improves pulmonary physiological function. The inflammatory response in WT (red) and *Esm1*^−/−^ (blue) mice after the intratracheal instillation of LPS at different time points. The evolution of endogenous endocan serum concentrations at different timings are shown in (**A**). The BAL examinations are shown, with the absolute number of leukocytes (**B**), neutrophils (**C**), and macrophages (**D**). mMPO BAL concentrations are shown in (**E**), mTNF BAL concentrations are shown in (**F**), and protein BAL concentrations are shown in (**G**). The evolution of transcutaneous oximetry at different timings is shown in (**H**). Data are represented by their median, 25th, and 75th percentiles. ∆ represents the estimation of the effect of the group according to the linear mixed model test. ɸ is the result of the Kruskal–Wallis test. * represents a *p* < 0.05 as the result of the Mann–Whitney test between the variable of each group at a specific timing. H12: 12 h after tracheal instillation of LPS.

**Figure 3 cells-12-00257-f003:**
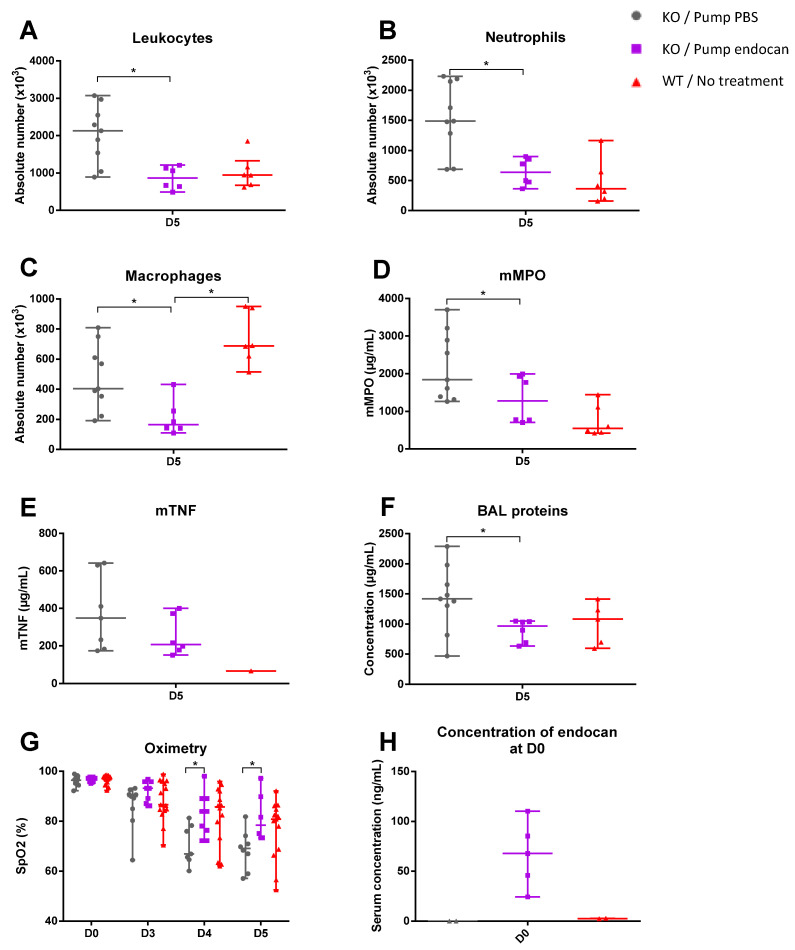
Continuous infusion of endocan reduces the inflammatory response and improves the pulmonary physiological function in *Esm1*^−/−^ mice. The inflammatory response on day 5 after intratracheal instillation of LPS in *Esm1*^−/−^ mice treated over 3 days by the continuous infusion of murine endocan (purple) compared to PBS (grey) and WT mice (red). BAL examinations are shown with absolute numbers of leukocytes (**A**), neutrophils (**B**), and macrophages (**C**). mMPO BAL concentrations are shown in (**D**), mTNF BAL concentrations are shown in (**E**), and protein BAL concentrations are shown in (**F**). Transcutaneous oximetry measurements at different times are shown in (**G**). The serum concentration of endocan at D0 is shown in (**H**). Data are represented by their median, 25th, and 75th percentile. ∆ is equal to the estimation of the effect of the group according to the linear mixed model test. * represents a *p* < 0.05 as the result of the Mann–Whitney test between the variables of each group at a specific timing.

**Figure 4 cells-12-00257-f004:**
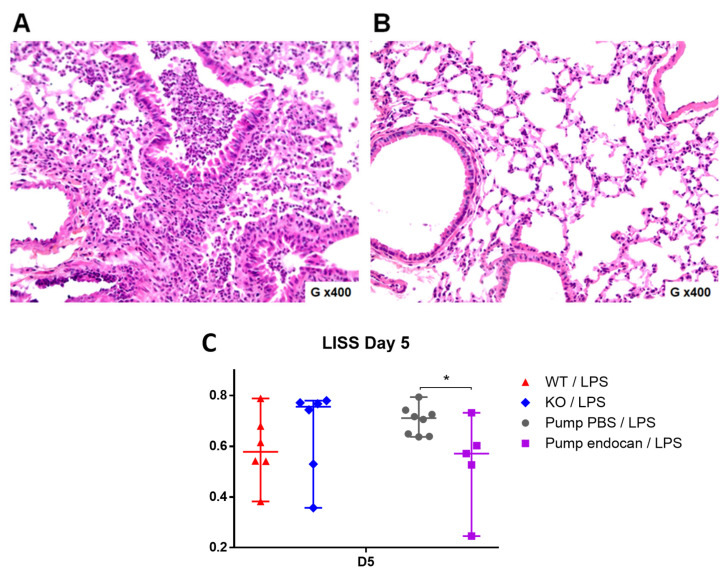
Continuous infusion of endocan alleviates the histological features of lung injury in *Esm1*^−/−^ mice. Histological examination of the lung and assessment with the lung injury severity score (LISS) at day 5 after intratracheal instillation of LPS in *Esm1**^−/−^* mice treated over 3 days by continuous infusion of PBS vs. murine endocan ((**A**), 200×; (**B**), 200×). (**A**), we present the histological photography of an *Esm1*^−/−^ mouse treated with a continuous infusion of PBS. We can observe patchy neutrophilic infiltrates, hyalines membranes, and fibrin. (**B**), the histological photography of an *Esm1*^−/−^ mouse treated with a continuous infusion of murine endocan is shown. (**C**), we present the results of LISS between the WT (red) and the *Esm1*^−/−^ (blue) groups and between the *Esm1*^−/−^ group treated either with PBS (grey) or endocan (purple). Data are represented by their median, 25th, and 75th percentile. * represents *p* < 0.05 as the result of the Mann–Whitney test between the variables of each group.

## Data Availability

The data presented in this study are available on request from the corresponding author.

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
