# Peer review of "Evaluation of Endocan as a Treatment for Acute Inflammatory Respiratory Failure"

_cells, 2023, doi:10.3390/cells12020257_

Round 1

Reviewer 1 Report (Previous Reviewer 2)

My concersn were addressed in the revised version.

I dont have any further cooments.

Author Response

My concersn were addressed in the revised version.

I dont have any further cooments.

Thank you for your helpful reviewing and comments.

Reviewer 2 Report (Previous Reviewer 1)

I thank the authors for making appropriate changes that helped improve their manuscript.

Some minor comments:

Abstract:

1. The aim of this study was to evaluate the efficiency of endocan in the treatment of ARDS.

Please correct: To evaluate the potential efficiency. There is no perfect animal model for ARDS (and usually 2-hit models are preferred).

2. Results, p6, line 241: was more important. Please re-phrase, more pronounced, maybe?

3. Endogen endocan. Please correct, endogenous endocan.

Author Response

I thank the authors for making appropriate changes that helped improve their manuscript.

Some minor comments:

Abstract:

  1. The aim of this study was to evaluate the efficiency of endocan in the treatment of ARDS.

Please correct: To evaluate the potential efficiency. There is no perfect animal model for ARDS (and usually 2-hit models are preferred).

We changed that sentence as suggested by the reviewer.

  1. Results, p6, line 241: was more important. Please re-phrase, more pronounced, maybe?

We replaced “more important” by “more pronounced” as suggested.

  1. Endogen endocan. Please correct, endogenous endocan.

We replaced “endogen” by “endogenous” throughout the manuscript.

Thank you for your helpful reviewing and comments.

This manuscript is a resubmission of an earlier submission. The following is a list of the peer review reports and author responses from that submission.

Round 1

Reviewer 1 Report

In this manuscript the authors' main conclusion is that endocan supplementation after LPS-induced ALI in Esm1-/- mice decreases lung inflammation and alleviates pulmonary physiological dysfunction, as demonstrated by a significant decrease in total leukocytes, neutrophils, and macrophages recruitments compared to those treated with PBS. 

I believe that important controls are missing:

I find this an expected finding. You manipulated animals to delete the ESM1 gene and then exogenously gave the product of the same gene to them.  

These experiments have to be performed in the WT LPS-induced ALI mice. If endocan is proposed as a treatment strategy in human ARDS, then it should also work in the WT mice. One cannot expect the patients to have both alleles of the ESM1 gene deleted. Hence, I believe that the animal model chosen is inappropriate due to the presence of the non-glycosylated form. ESM1-/- mice seem to exhibit a more severe LPS-induced ALI profile than the WT. So you have to compare ESM1-/- ALI+endocan versus WT control mice. To see if ALI is treated or you have just attenuated it.

Also, you have not compared the ESM1 knock out mice with WT mice.

In the ESM1 knock-out mice, leukocyte infiltration was increased in LPS-induced injury, as expected.

Figure 2: What is the time-point H12? Please explain both in the text and the Figure legend.

Results: The comparison between Esm1-/- mice and WT littermates showed that endogenous murine endocan decreased the severity of LPS-induced ALI and alleviated lung physiological dysfunction, as reflected by greater SpO2 values in the WT group.

Please re-phrase. May potentially. You do not show in Figure 2 that endocan decreases severity.

However, LISS values were not different between the two groups (Figure 4). This latter result may not be surprising since in mouse, the anti-inflammatory glycosylated endocan and its non-glycosylated competitive antagonist both coexist. Thus, the knock-out of both molecules may not induce major changes in LISS. The difference observed in the cellularity in BAL fluids and SpO2 suggest that these parameters are more sensitive than LISS.

Exactly, so how can you draw such conclusions on the point above?

Moreover, continuous infusion of glycosylated murine endocan in Esm1-/- mice reduced the histological features of LPS-induced ALI, as reflected by the lower LISS values observed in treated mice.

But LISS values were not different between WT-LPS and ESM1-/- LPS (Figure 4).

Reviewer 2 Report

In the brief manuscript entitled ‘Evaluation of endocan as a treatment for acute inflammatory 2 respiratory failure’ by Hureau et al., show that delivery of glycosylated endocan in murine model of LPS-induced acute lung injury protects from lung injury.

I don’t have major questions, but some minor concerns are as follows

1.      What are endocan-1 levels in this ALI model

2.      Macrophage numbers in endocan treated mice at d5 and d6 are opposite to each other. Are there any changes in M1 vs M2 subsets as well?

3.      Changes in vascular density and vascular leakage in endocan treated mice are necessary to understand the changes in immune cell infiltration/numbers

4.      Did the authors check whether there are any changes in non-glycosylated vs. glycosylated endocan at the end of study in glycosylated endocan treated mice